# Corneal Nerve Changes Observed by In Vivo Confocal Microscopy in Patients Receiving Oxaliplatin for Colorectal Cancer: The COCO Study

**DOI:** 10.3390/jcm11164770

**Published:** 2022-08-16

**Authors:** Ellen F. Tyler, Charles N. J. McGhee, Benjamin Lawrence, Geoffrey D. Braatvedt, Joseph L. Mankowski, Jonathan D. Oakley, Sargun Sethi, Stuti L. Misra

**Affiliations:** 1Department of Ophthalmology, New Zealand National Eye Centre, Faculty of Medical and Health Sciences, The University of Auckland, Auckland 1142, New Zealand; 2Department of Oncology, The University of Auckland, Auckland 1142, New Zealand; 3Department of Endocrinology, Greenlane Clinical Centre, Auckland District Health Board, Auckland 1051, New Zealand; 4Department of Molecular and Comparative Pathobiology, Johns Hopkins University, Baltimore, MD 21205, USA; 5Voxeleron LLC, 13809 Research Blvd., Suite 500, Austin, TX 78750, USA

**Keywords:** chemotherapy-induced peripheral neuropathy, corneal nerves, in vivo confocal microscopy of cornea, corneal sensitivity

## Abstract

An objective method of early identification of people at risk of chemotherapy-induced peripheral neuropathy is needed to minimize long-term toxicity and maximize dose intensity. The aims of the study were to observe corneal nerve microstructure and corneal sensitivity changes and peripheral neuropathy in patients receiving oxaliplatin, and to determine its association with corneal parameters at different stages of treatment and assess utility as non-invasive markers to detect and monitor peripheral neuropathy. Twenty-three patients scheduled to receive oxaliplatin chemotherapy with intravenous 5-FU for gastro-intestinal cancer were recruited and followed up with for 12 months. Ocular examinations including corneal and retinal evaluations, alongside peripheral neuropathy assessment, were performed. The corneal nerve density did not show significant change after chemotherapy when measured with a widely used semi-automated program or an automated analysis technique. Macula and optic nerve function did not change during or after oxaliplatin chemotherapy. However, the corneal nerve density modestly correlated with clinical peripheral neuropathy after 20 weeks of chemotherapy (r = 0.61, *p* = 0.01) when peripheral neuropathy is typical most profound, and corneal nerve sensitivity correlated with neuropathy at 12 (r = 0.55, *p* = 0.01) and 20 weeks (r = 0.64, *p* = 0.006). In conclusion, corneal changes detected on confocal microscopy show moderate association with peripheral neuropathy, indicating their potential to identify the development of oxaliplatin-induced peripheral neuropathy. However, further studies are required to confirm these findings.

## 1. Introduction

Oxaliplatin and cisplatin can cause severe peripheral sensory neuropathy [1]. Oxaliplatin chemotherapy is in widespread use across the world, coupled with either 5-fluorouracil or oral capecitabine, as the standard of care for many gastrointestinal cancers in both curative and palliative settings and improves survival [2,3,4]. Chemotherapy-induced peripheral neuropathy negatively affects these patients’ quality of life, increases health care costs [5] and is the most common dose-limiting toxicity of oxaliplatin [2].

An objective prospective method of assessing chemotherapy-induced peripheral neuropathy is needed to minimize long-term toxicity and maximize dose intensity. Corneal in vivo confocal microscopy (IVCM) is a unique non-invasive tool that allows the assessment of small sensory fibres by direct observation of corneal nerve microstructure. Recently, it was shown that corneal IVCM and corneal sensitivity threshold (CST) assessment are sensitive clinical tools to detect early diabetic peripheral neuropathy and may be clinically used to diagnose and monitor the progression of neuropathy [6,7,8,9,10]. Therefore, it may be possible to use corneal IVCM and CST as surrogate, or even prognostic, markers for chemotherapy-induced peripheral neuropathy. Two separate studies, comprising 13 and 15 participants, respectively, showed that peripheral neuropathy and corneal nerve changes co-exist during or after chemotherapy [11,12]. Recently, a cross-sectional study reported reduced corneal nerve density in patients with breast or gynaecological cancer who had been treated with paclitaxel, and a case report revealed a high number of corneal dendritic cells after paclitaxel treatment [13,14]. A number of atypical bulb-like enlarged nerve endings, termed neuromas, were also detected in the corneal sub-basal nerve plexus of patients treated for multiple myeloma [13,14,15]. Despite evidence of corneal nerve dysfunction developing in patients receiving chemotherapy in different type of cancers, limited longitudinal studies exist assessing these changes over time.

Dose intensity of oxaliplatin chemotherapy is positively associated with colorectal cancer survival, but unfortunately, there is also a clear association between the total cumulative dose and the development of chronic peripheral neuropathy. Whereas oncologists decrease the dose delivered or terminate oxaliplatin therapy in response to the development of peripheral neuropathy, the dose adjustment often occurs too late to prevent chronic permanent peripheral neuropathy [16].

Currently, there are no objective methods by which clinicians can predict the onset of chronic peripheral neuropathy in patients receiving chemotherapy, nor identify those at most risk. A recent study shows that corneal changes can be detected up to 2 years after oxaliplatin-based chemotherapy [17], but limited literature exists for prospective, longitudinal studies investigating potential chemotherapy-induced corneal abnormalities temporally associated with developing peripheral neuropathy. IVCM could potentially provide a more refined, objective surrogate for the detection of oxaliplatin-related peripheral neuropathy. The overall aim of the study was to study corneal nerve microstructure changes observed by in vivo confocal microscopy in patients receiving oxaliplatin (COCO), and secondly, to assess oxaliplatin-induced peripheral neuropathy and determine its correlation with corneal nerve microstructure and corneal sensitivity at different stages of treatment. If changes in nerve microstructure are detected in the cornea, and these changes are associated with the development of chemotherapy-induced neuropathy, there is potential to utilise such changes as non-invasive markers in the detection and monitoring of oxaliplatin-induced peripheral neuropathy.

## 2. Materials and Methods

The current study is part of a larger research project approved by the local ethics committee (NTX/11/06/051/AM02) and adheres to the tenets of the Declaration of Helsinki. All subjects gave written informed consent before participating in the study. Annually, approximately 200 patients are treated using oxaliplatin in Auckland Hospital, and the drug has been reported to cause sensory neuropathy in up to 80% of these patients [18]. Attempting to calculate a sample size based on repeated measures analysis was difficult due to the novel nature of the study. At present, there is no estimate for the value of the mean corneal nerve density (the primary outcome measure) at each time point, nor is there a value for the correlation between each time point, and these would be needed to calculate sample size. Hence, this research was defined as a longitudinal observational study without any need of sample size calculation. All the patients undergoing oxaliplatin treatment for gastrointestinal cancer who meet the criteria will be requested to participate. This observational study will gather the data and values needed to form the foundation for a detailed sample size calculation for future randomized trials.

Patients commencing oxaliplatin chemotherapy with intravenous 5-FU for any gastro-intestinal cancer were recruited. Patients with history of any bilateral ocular surgery, current contact lens wear or any systemic disease causing peripheral neuropathy, were excluded. All patients were recruited under the supervision of a senior medical oncologist (BL). Patients underwent ophthalmologic assessments at baseline (prior to first dose of oxaliplatin) and at six, 12, 20, 36 and 52 weeks after the first chemotherapy session. In addition, each patient was requested to undergo standard peripheral nerve electrophysiology at baseline and again at the 20-week follow up.

The external and anterior ocular structures were examined using a Topcon slit lamp biomicroscope (Topcon Medical Systems, Oakland, NJ, USA). CST was evaluated with the Non-Contact Corneal Aesthesiometer [19]. Assessment of colour vision deficiency was performed with Ishihara test plates. Colour vision assessment reflects optic nerve function and deficiencies can indicate the presence of an optic neuropathy. Pupil responses were checked individually and then alternately in the swinging light test to detect RAPD (relative afferent pupillary defects). Abnormal pupillary responses indicate a potential optic neuropathy.

The central cornea of the right eye of each patient was examined at each visit with laser scanning in vivo confocal microscopy in accordance with a well-established protocol [6,20]. The cornea was anaesthetised with one drop of 0.4% benoxinate hydrochloride (Chauvin Pharmaceuticals Ltd., Kingston, UK). A drop of Viscotears liquid gel (Viscotears Carbomer, Novartis Pharmaceuticals Ltd., London, UK) was placed on the lens objective, avoiding bubbles and serving as a contact substance. A disposable Tomocap was then mounted over the objective lens of the confocal microscope. The captured images represent two-dimensional optical sections through the cornea at the location of the focal plane. These images were individually evaluated to calculate the density of the sub-basal nerve plexus. All images were collected, and all follow up visits were completed before any analysis of IVCM images was started. Images were sent to an independent observer (SM) and randomised through a computer-based random number generator. The randomised and de-identified images were subsequently traced by an experienced clinician for corneal nerve density (ET) and processed to determine the presence of neuromas or dendritic cells (SS). Visible nerves were traced with an electronic pen with a digital caliper tool (Wacom Technology Group, Vancouver, BC, Canada) and measuring the total length of nerves per frame using semi-automated software ImageJ with NeuronJ (National Institutes of Health, Bethesda, Rockville, MD, USA). This parameter is arguably the most repeatable and provides maximum information about corneal nerve microstructure [21]. To note, the terms sub-basal nerve density and sub-basal nerve length, measured as mm/mm^2^, are used interchangeably in the literature due to lack of unanimous consensus in the terminology [6,20,22,23,24].

Automated analysis of corneal nerves was also performed to complement semi-automated assessments. A customised deep learning-based approach named deepNerve has been recently developed to objectively measure corneal nerve density or corneal nerve fibre length—the total length of the nerves divided by the image area, expressed as mm/mm^2^—as well as secondary metrics such as tortuosity [25]. Initially validated using the manually traced IVCM images of macaque cornea, the approach is now used in studies of pre-clinical animal models [25,26]. The output from the network is a nerve probability map where pixels are in the range 0, indicating no nerves, to 1, where nerves are noted [25]. A final thresholding of this result creates a binary image from which quantitative parameters are derived.

Optical Coherence Tomography (OCT) is a standard non-invasive assessment of the retina based on a coherent light source and principles of interferometry to capture high resolution, cross-sectional images. At every visit, OCT was performed on the patient’s right retina and optic nerve to assess any retinal changes or optic neuropathy. The specific OCT scan used was a nine-line raster cross of the macula, to assess for potential Oxaliplatin maculopathy.

The Total Neuropathy Score (TNS) was calculated by combining a three-part assessment: questionnaire, biothesiometry and clinical assessment. The first part involved filling in a “symptoms of neuropathy” questionnaire that evaluates and grades the severity of sensory symptoms such as numbness, pain and sensitivity in the extremities; the second part comprised a peripheral nervous system assessment including biothesiometry by an experienced clinician (ET); and the third part included testing the vibration threshold of the nerves of the feet using a bioesthesiometer—ankle and knee reflexes and big toe proprioception as well as the ten-gram monofilament test [27]. The score ranged between 0 to 30, and the higher the score, the worse the neuropathy. A subset of patients underwent a baseline nerve conduction study (NCS) and a 20-week follow up review. This is a measurement of sural and peroneal conduction velocity and amplitude [27]. The study determines myelinated nerve function by preferentially assessing the fastest conducting subset of the alpha motor axon population. A reduction in conduction velocity or amplitude is indicative of neuropathy. A score is calculated between 0 to 10, with 0 being no neuropathy and 10 being the worst case scenario. NCS scores were not included in the TNS due to limited (eight) patient data at baseline and at 20-week assessment. The Kolmogronov–Smirnov test was performed to confirm normality of the data distributions. Multiple imputation methodology was administered to process the missing data due to missed assessments [28]. Repeated measures ANOVA was administered to assess changes due to oxaliplatin over the one-year period, with and without employing imputation models. *p <* 0.05 was considered significant. SPSS Statistics 23 (IBM Corporation, Armonk, NY, USA) was used for analysis. Data are presented as mean ± SD.

## 3. Results

Twenty-three patients (15 males:8 females) were recruited over a 12-month recruitment period. The mean age of patients was 61.0 ± 11.0 years (mean ± standard deviation) at baseline, and 19 were Caucasian, three were Asian and one was New Zealand Māori.

The median oxaliplatin cumulative dose was 762.5 mg (range 150 to 1780 mg) across the three cycles. All patients had intra-venous 5FU. Twenty-three patients were assessed at the baseline visit—20 at 6 weeks, 22 at 12 weeks, 19 at 20 weeks, 6 at 36 weeks and 11 at the final 52-week visit.

At baseline, the corneal sub-basal nerve density was 15.9 ± 6.2 mm/mm^2^ (mean ± standard deviation), shown in Figure 1 and Table 1. The box and whisker plot (Figure 2) did not show a change in corneal nerve density over the course of the study. Similarly, no changes were noted after applying an imputation model with the worst case imputation scenario Wilk’s Lambda = 0.65, F(5,18) = 1.82, *p* = 0.16, n^2^ = 0.37. This included the 20-week assessment conducted while receiving chemotherapy and the follow up at 52 weeks to capture any change after cessation of treatment.

The corneal nerve density measurement with semi-automated tracing software NeuronJ and deepNerve automated analysis showed excellent comparability (r^2^ = 0.88, Figure 3). Table 2 summarises the corneal nerve density of each patient at each follow-up assessment, as well as the final patient outcome after the follow up time frame.

The corneal sensitivity threshold was 0.3 ± 0.1 mBAR at the baseline visit. Table 1 presents the means of the corneal sensitivity threshold as measured by NCCA, along with the number of patients showing the presence of neuropathy, RAPD or any retinal abnormalities.

Out of 23 patients, nine patients had corneal neuromas and 11 had corneal dendritic cells at baseline, including two who had both. During and after the treatment conclusion, a further 14 had neuromas and 11 had dendritic cells. Amongst these patients, ten developed both dendritic cells as well as neuromas. All of these patients except one exhibited some degree of peripheral neuropathy during or after chemotherapy. Representative examples of neuromas and dendritic cells are shown in Figure 4.

No macular abnormalities were documented on baseline OCT scans and no change in OCT appearances was noted in any patient during this study. No patient had RAPD at any of the assessments, and no patient reported any change in their colour vision as assessed by Ishihara colour plates. Two patients had colour blindness (inherited red-green colour anomaly) that did not change at any stage of the study.

Peripheral neuropathy was assessed using the TNS and the electrophysiological NCS. Out of 23, six patients had minimal neuropathy detected at the baseline visit by NCS, whereas TNS (questionnaire, clinical assessment and biothesiometry) did not suggest neuropathy. During chemotherapy at the 20-week assessment, 16 out of 17 had some degree of neuropathy detected by TNS or NCS (Table 1). Only three patients out of 11 available for follow up had some level of neuropathy at the concluding 52-week visit. Ten patients underwent nerve conduction study at baseline, whereas only nine were re-assessed at 20-week time-point (Table 1). The peroneal distal compound muscle action potential (*p* = 0.93) and sural sensory nerve action potential amplitude (*p* = 0.1) did not change from baseline to the 20-week timepoint. These amplitudes did not correlate with corneal nerve density; however, this lack of association should be interpreted with caution due to lower number of participants with NCS. All had normal TNS at the baseline visit, which showed a minor change at 20-week timepoint.

There was no association between corneal nerve density, peripheral neuropathy as indicated by total neuropathy score and cumulative oxaliplatin dosing (Table 3) in most of the timepoints, apart from the 20-week assessment, which revealed a moderate association between corneal nerve density and cumulative dosing (r = 0.61, *p* = 0.01). There was no trend or association between neuroma or dendritic cell development with chemotherapy cycles. However, there was moderate correlation between corneal sensitivity and the total neuropathy score at the 12- (r = 0.55, *p* = 0.01) and 20-week (r = 0.64, *p* = 0.006) timepoint (Table 3).

## 4. Discussion

The current prospective, longitudinal study did not detect changes in the corneal density, macula or optic nerve during or after oxaliplatin chemotherapy, in contrast to previous studies that showed significant oxaliplatin related ocular surface changes [11,12]. This study, however, did reveal a moderate association between corneal sensitivity and total neuropathy score, suggesting functional changes in the cornea that are not mirrored in corneal nerve density. The inconsistencies between the studies could potentially be due to discontinuation of follow up following chemotherapy cessation in previous studies.

The association between corneal changes and peripheral neuropathy at 20 weeks corresponds with the usual timepoint that peripheral neuropathy becomes severe in the oncology clinic, suggesting that both corneal nerve density and corneal sensitivity threshold are affected at the same time. Interestingly, peripheral neuropathy occurring after 12 weeks of oxaliplatin is usually mild, so the association of peripheral neuropathy with corneal sensitivity at this early timepoint (and again at 20 weeks) suggests corneal sensitivity might be a more sensitive measure than corneal sub-basal nerve density. There is possible merit in assessing the corneal sensitivity threshold in patients undergoing chemotherapy to monitor peripheral neuropathy at various stages of their treatment.

Oxaliplatin has been associated with several ocular side effects. Several studies report the common occurrence of dry eye, epiphora, eye pain and ptosis associated with the acute phase of neurotoxicity and rare cases of reduced visual acuity, optic neuropathy, visual changes and visual loss [17,29,30,31]. Patients in the current reported study frequently reported symptoms of dry eye and associated eye discomfort (often associated with 5-flourouracil); however, none of the patients experienced any visual loss, visual changes or optic neuropathy associated with oxaliplatin. Painful dry eye disease is associated with corneal nerve microstructural changes [32]. Further research is needed to investigate the role of dry eye associated corneal nerve in patients undergoing chemotherapy. Furthermore, optic nerve function testing, visual acuity and OCT scans through the macula remained unchanged throughout the follow up period.

Of note, the measurement of corneal nerve density using the conventional manual tracing and novel automated deepNerve methodology produced almost identical outcomes in the current study. The deepNerve methodolgy has previously been used in animal as well as human studies [25,26,33]. The corneal nerve density or corneal nerve length measured as mm/mm^2^ is a reproducible parameter reported to reduce in diseases often associated with peripheral neuropathy, for example with diabetes and Parkinson’s disease [21,22,23,24,34].

Previous case series observed significant corneal changes, as imaged by IVCM, in patients with peripheral sensory neuropathy caused by oxaliplatin-based chemotherapy treatment [11,35]. In a case-control series, Campagnolo et al. recruited fifteen patients receiving oxaliplatin chemotherapy, assessing them prior to commencement of chemotherapy and after completion of chemotherapy [11]. They were matched with sixteen healthy age-matched subjects. Investigations at both time points included corneal IVCM, corneal aesthesiometry TNS, NCS, OCT and pupillometry [11]. Out of the fifteen oxaliplatin treated patients in the Campagnolo et al. study, ten had reduced TNS at the end of chemotherapy and eight had clinical signs of neuropathy [11]. At the same timepoint in the current study, a similar proportion of patients (17 of 19) had peripheral neuropathy.

The translational aim of the current study was to determine whether corneal changes could be used as an early biomarker for subsequent permanent neuropathy. Two of the four patients with IVCM changes but stable TNS scores in the Campagnolo et al. study developed clinical peripheral neuropathy several weeks after completing their final cycle of chemotherapy, suggesting the IVCM has potential in identifying patients who are likely to experience the coasting effect of oxaliplatin neuropathy [11]. Although corneal nerve microstructure was associated with peripheral neuropathy at 20 weeks in the current study, notably, corneal sensitivity was correlated with the total neuropathy score. We were not able to determine an association with later permanent neuropathy because it was rare at the 52-week mark; in fact, peripheral neuropathy was detected at lower levels than at baseline. Patient attrition also possibly played a role.

A recent cross-sectional study by Chiang et al. showed a difference in sub-basal nerve length, as measured by mm/mm^2^, in 29 patients with breast or gynaecological cancer (14.2 mm/mm^2^ ± 4.0) compared to control participants (16.4 mm/mm^2^ ± 4.0) [13]. The current study noted 15.9 mm/mm^2^ ± 6.2 at the baseline visit (23 patients) and 15.5 mm/mm^2^ ± 4.8 at the 20-week visit (17 patients), whereas the Chiang et al. patient cohort had a history of chemotherapy between 3 to 24 months prior to ocular assessment [13]. Another study by Chiang et al. showing significant differences in oxaliplatin-induced peripheral neuropathy and oxaliplatin-related ocular toxicity between patients who underwent chemotherapy, and control participants without cancer or chemotherapy is clearly unsurprising [17]. However, a study of thirteen patients did exhibit changes in corneal nerve microstructure after three cycles of oxaliplatin or cisplatin. Interestingly, a study of 13 patients with gastro-intestinal cancer showed re-generation of corneal nerves after three cycles of oxaliplatin, epirubicin, cisplatin or capecitabine [12]. It is imperative to assess patients before, during and after cessation of treatment to draw firm conclusions. Our study did assess patients at various timepoints, but patient attrition due to morbidity and mortality has the potential to bias the results.

The presence of dendritic cells and neuromas in patients with cancer and chemotherapy warrants further longitudinal studies with larger patient cohorts. The dendritic cells migrate from the peripheral corneal towards the centre during an inflammatory state [36,37]. Although the current study did observe dendritic cells in 15/23 patients over the course of chemotherapy, there was no trend or association with treatment time-line. In this study, neuromas were noted in 18/23 patients, and these were noted in a previous study in 18/25 of patients with multiple myeloma [15]. Neuromas are described as anomalous nerve features that are often associated with corneal neuropathy and ocular surface disease [38,39]. However, there is no consensus amongst researchers regarding the importance and relevance of these structures due to lack of standard neuroma definition in the literature [40].

In comparison with the preceding published literature, the current COCO study has the longest follow up period, wherein the patients were assessed before, during and after chemotherapy. Interestingly, at most time points, no statistically significant correlation was identified between cumulative doses of oxaliplatin and corneal nerve density or between oxaliplatin-induced peripheral neuropathy and corneal nerve density, except at the time of most peripheral nerve damage, and a moderate association between corneal nerve density and cumulative dosing was noted (r = 0.61, *p* = 0.01). The correlations are suggestive, but at an individual level, some patients showed a decrease in corneal nerve density, where other did not, suggesting that IVCM corneal nerve density is currently an insensitive method of detecting oxaliplatin-induced peripheral neuropathy. Yet, the association between corneal sensitivity and total neuropathy score does suggest potential parallel changes in the cornea and peripheral extremities, introducing a potential confounding variable into studies of neuropathy in the treatment of cancers.

Interestingly, corneal nerve fibre length and density were reduced in patients with upper gastrointestinal cancer without chemotherapy in comparison to age-matched controls, which was attributed to systemic peripheral neuropathy due to cancer [12]. The current study also noted lower sub-basal nerve density prior to commencement of chemotherapy.

A limitation of the current study was slow recruitment and loss of patients during the study. In addition to lower-than-expected numbers of eligible patients presenting to the new patient clinics, an atypically low uptake was experienced in the recruitment period. Indeed, of the 117 potentially eligible patients, only 23 patients agreed to participate, whereas an additional four patients initially consented to the study but withdrew their consent before the baseline examination. Several patients who did not consent to the study contacted the investigator following commencement of chemotherapy, expressing interest but could not be included as no baseline data could be established. Of the 23 patients recruited, only ten patients (39%) consented to additional nerve conduction testing, two of whom did not undergo their second testing at 20 weeks.

Overall, patients expressed anxiety at the initial recruitment approach at the Oncology New Patient clinic, and although most patients were already aware of their diagnosis of cancer, this was their first meeting with an oncologist. Understandably, this is an emotional time for the patient and their family members, and a lot of information is presented in a short space of time. There was then attrition during the study despite considerable effort made to make the ophthalmic and nerve conduction testing as convenient and straightforward as possible to attend.

Oxaliplatin-induced peripheral neuropathy is related to the cumulative dose of oxaliplatin; the higher the cumulative dose, the greater the risk of developing neuropathy [41]. Unfortunately, the only effective method of preventing oxaliplatin-induced peripheral neuropathy is to reduce the dose of oxaliplatin delivered or terminate treatment. The COCO study did not identify significant changes in corneal nerves during oxaliplatin chemotherapy, found only moderate correlations between peripheral neuropathy and corneal innervation and could not assess the potential of corneal innervation as a surrogate for peripheral neuropathy. At this stage, it cannot be suggested that corneal nerve density is a reliable surrogate biomarker for oxaliplatin neuropathy. Subsequent larger studies may be able to more clearly define any role that IVCM of corneal nerve density may play in the assessment of chemotherapy-induced peripheral neuropathy. Further investigations including corneal sensitivity measurements are required to conclude whether its moderate association with peripheral neuropathy occurs in tandem.

## 5. Conclusions

The current study suggests that corneal nerve density may not be an effective biomarker for oxaliplatin-induced peripheral neuropathy; however, further studies are required to confirm these findings.

## Figures and Tables

**Figure 1 jcm-11-04770-f001:**
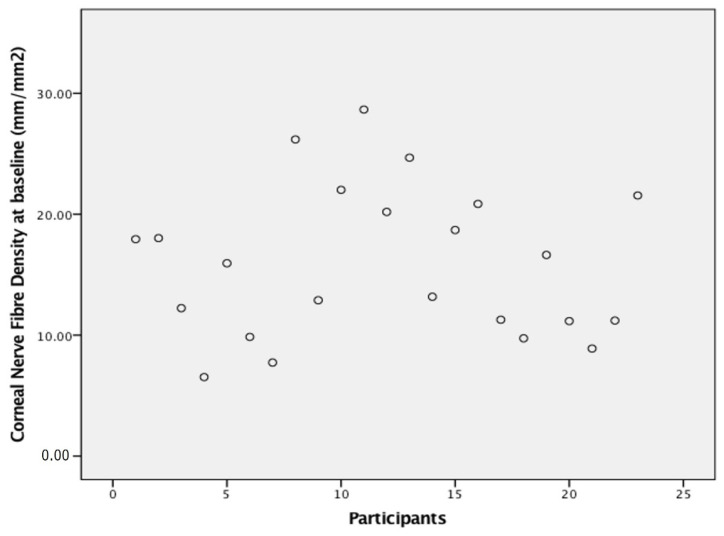
Scatterplot of corneal nerve fibre density of participants at baseline assessment showing the spread of corneal nerve density at baseline. Note a group of patients with a baseline density below 15 mm/mm^2^.

**Figure 2 jcm-11-04770-f002:**
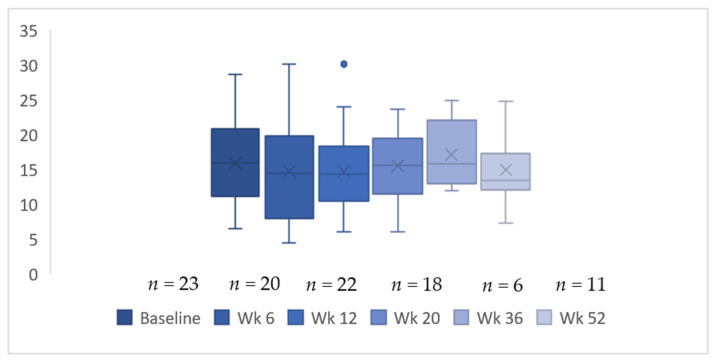
Box and whisker plot of change in corneal nerve fibre density over time. Whiskers represent 95% confidence intervals.

**Figure 3 jcm-11-04770-f003:**
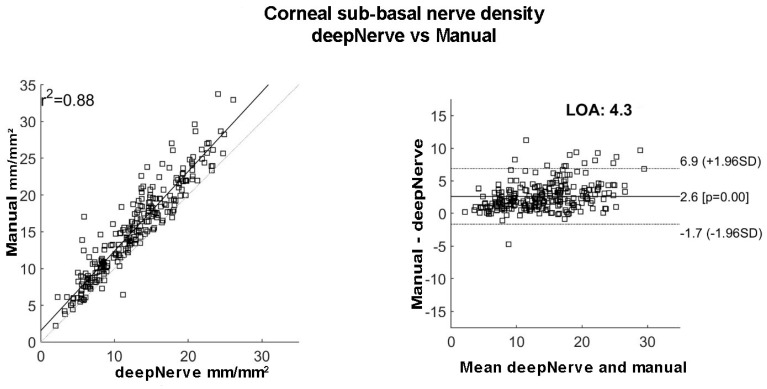
(**left**) comparability of sub-basal nerve density measurement using deepNerve and manual tracing expressed as mm/mm^2^; (**right**) Bland and Altman plot to compare manual tracing and deepNerve to measure sub-basal nerve density.

**Figure 4 jcm-11-04770-f004:**
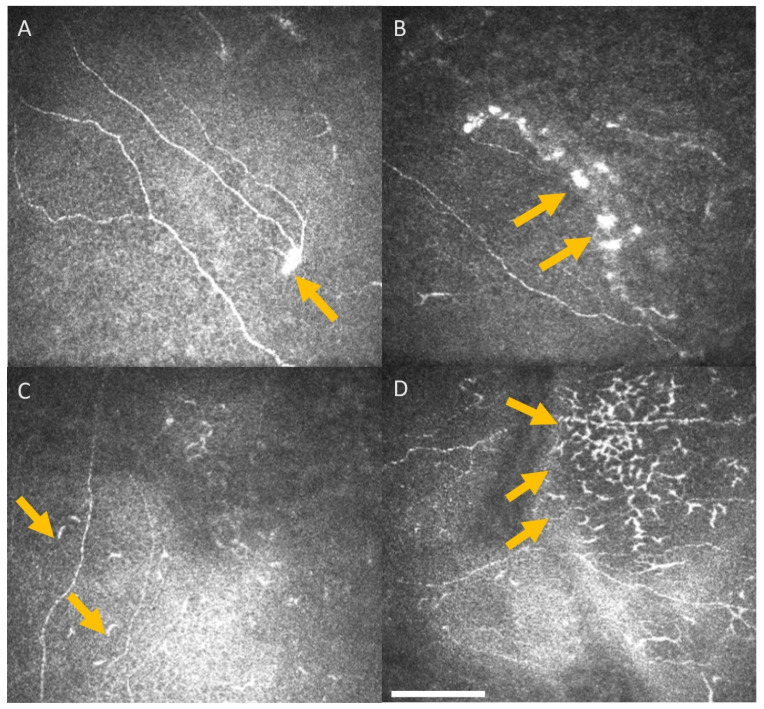
Representative central corneal IVCM images. Arrows highlight neuromas (**A**,**B**) and dendritic cells (**C**,**D**), identified on the images obtained from the central cornea sub-basal nerve plexus of patients undergoing oxaliplatin chemotherapy for colorectal cancer. Scale bar = 100 µm.

**Table 1 jcm-11-04770-t001:** Means of corneal nerve density and corneal sensitivity threshold at each visit with standard deviations noted. The presence of dendritic cells, neuromas and neuropathy, alongside total neuropathy score and nerve conduction study (NCS) scores are along with the number of patients with Relative Afferent Pupillary Defect (RAPD) and retinal abnormalities are shown. NCS was only done at baseline and at 20 weeks. *n* = number of patients.

		Baseline	6 Weeks	12 Weeks	20 Weeks	36 Weeks	52 Weeks
		*n* = 23	*n* = 20	*n* = 22	*n* = 18	*n* = 6	*n* = 11
Corneal sub-basal nerve density		15.91 ± 6.2	14.73 ± 6.8	14.74 ± 5.8	15.51 ± 4.8	17.51 ± 4.9	14.91 ± 4.9
Presence of dendritic cells (*n*)		11	3	7	4	1	4
Presence of neuromas (*n*)		9	10	6	8	3	3
Corneal sensitivity threshold		0.39 ± 0.1	0.38 ± 0.2	0.49 ± 0.3	0.47 ± 0.3	0.25 ± 0.3	0.19 ± 0.3
Presence of peripheral neuropathy (*n*) (NCS or TNS)		6	16	17	16	3	3
TNS score (0–30)	Median	0	1	3	5	3.5	0
	Range	0	0–7	0–7	0–13	0–13	0–14
NCS (0–10)	Median	0			6		
	Range	0 (*n* = 10)			0–6 (*n* = 9)		
Presence of RAPD (*n*)		0	0	0	0	0	0
Retinal abnormalities (*n*)		0	0	0	0	0	0

**Table 2 jcm-11-04770-t002:** The corneal nerve density (CND) of each patient at each follow-up assessment and the final patient outcome at the completion of the follow-up time frame and reasons for dropout. Patients 1, 6, 8, 13, 15 and 17 showed a decrease (light grey), whereas patients 4, 5, 7, 18 and 22 showed an increase (dark grey) in corneal nerve density.

	Baseline CND	6 Weeks CND	12 Weeks CND	20 Weeks CND	36 Weeks CND	52 Weeks CND	Patient Outcome
Patient 1	17.94	11.98	12.47				Deceased prior to 20 weeks
Patient 2	18.02	20.62	15.36	19.84			Disease progression at 20 weeks and referred to hospice
Patient 3	12.23	12.18	8.99	14.06	11.91	12.01	Completed follow up
Patient 4	6.53	4.43	6.61	10.26			Disease progression at 20 weeks Rx 2nd line treatment. Lost to follow up
Patient 5	15.95	19.83	16.59	20.05	24.91	24.82	Completed follow up
Patient 6	9.85	5.82	6.04				Moved overseas and deceased
Patient 7	7.73	14.49	14.49	15.41	13.38	13.38	Completed follow up
Patient 8	26.19	20.78	14.16	17.31	21.1	13.33	Completed follow up
Patient 9	12.89	12.96	12.96	14.61		12.88	Completed follow up
Patient 10	22.02	17.82	17.06	14.82	16.37	16.37	Completed follow up
Patient 11	28.66	30.11	30.11				Had a stroke, chemotherapy terminated and too unwell to attend clinics
Patient 12	20.19		18.73				Disease progression at 20 weeks and Rx 2nd line treatment. Lost to follow up
Patient 13	24.67	15.34	9.13	11.92			Offered resection of metastases at 20 weeks, no response to contact thereafter
Patient 14	13.18		11.68				Too unwell to attend
Patient 15	18.7	17.52	19.8	17.28		9.9	Completed follow up
Patient 16	20.86	23.96	23.96	22.17			Lost to follow up after completion of Rx
Patient 17	11.27	6.77		5.99			Lost to follow up after completion of Rx
Patient 18	9.74		10.94	8.99	15.25	17.3	Completed follow up
Patient 19	16.63	13.8	20.28	19.34		20.34	Completed follow up
Patient 20	11.16	17.39	15.64	15.64		16.39	Completed follow up
Patient 21	8.89	6.23	13.28	9.31		7.29	Completed follow up
Patient 22	11.21	7.92	7.92	18.63			Lost to follow up after completion of Rx
Patient 23	21.55		18.1	23.61			Lost to follow up after completion of Rx

**Table 3 jcm-11-04770-t003:** Spearman’s rho correlation between corneal nerve density vs. total neuropathy score and corneal nerve density vs. cumulative dose of oxaliplatin was computed. The correlation analysis was not possible at 36 weeks due to missing data. Asterisk * indicates statistically significant results.

		Baseline	6 Weeks	12 Weeks	20 Weeks	52 Weeks
		*n* = 23	*n* = 20	*n* = 22	*n* = 18	*n* = 11
Corneal nerve density vs. TNS	r	−0.04	−0.4	0.15	−0.1	−0.48
	*p*	0.83	0.09	0.54	0.69	0.13
Corneal sensitivity vs. TNS	r	0.12	0.34	0.55 *	0.64 *	0.05
	*p*	0.57	0.17	0.01 *	0.006 *	0.88
Corneal nerve density vs.	r		0.08	0.1	0.61	0.12
Cumulative dose of oxaliplatin	*p*		0.74	0.69	0.01	0.72

## Data Availability

Data is available upon request.

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
