# Peer review of "Corneal Nerve Changes Observed by In Vivo Confocal Microscopy in Patients Receiving Oxaliplatin for Colorectal Cancer: The COCO Study"

_jcm, 2022, doi:10.3390/jcm11164770_

Round 1

Reviewer 1 Report

The topic of the work is very relevant and useful. Sufficiently high interest was caused by the correlations that were found in the work. The authors in the work indicate references to works 20 and 21, which determine the length and tortuosity of the corneal nerve fibers. Why are data on the length of the main nerve fibers of the cornea and their tortuosity not presented in the work? Information about length and tortuosity would complement the text and be up to date. Thank you so much. Best wishes.

Author Response

  1. The topic of the work is very relevant and useful. Sufficiently high interest was caused by the correlations that were found in the work. The authors in the work indicate references to works 20 and 21, which determine the length and tortuosity of the corneal nerve fibers. Why are data on the length of the main nerve fibers of the cornea and their tortuosity not presented in the work? Information about length and tortuosity would complement the text and be up to date. Thank you so much. Best wishes.

Comment: We sincerely thank the reviewer for such positive and encouraging comments.

Reply: The sub-basal nerve density was evaluated by tracing visible nerves with an electronic pen (Wacom Technology Group, Vancouver, BC, Canada) and measuring the total length of nerves per frame with a digital caliper tool (ImageJ with NeuronJ plugin).[1-3] It should be noted that the terms sub-basal nerve density and sub-basal nerve length, measured as mm/mm2 are used interchangeably in the literature as there is no consensus regarding the terminology.[4-8] McCarron et al and Oakley et al., determine the length of the nerves measured as mm/mm2, as is the case in the current study.[9,10] Although several parameters, including nerve length tortuosity, are used to quantify corneal nerve microstructure,   the corneal nerve density or nerve length, measured as mm/mm2 is arguably the most repeatable.[11]

Action: The following sentences are amended/added in the methods section for clarity:

'The randomised and de-identified images were subsequently traced by an experienced clinician for corneal nerve density (ET) and processed to determine the presence of neuromas or dendritic cells (SS). Visible nerves were traced with an electronic pen with a digital caliper tool (Wacom Technology Group, Vancouver, BC, Canada), and measuring the total length of nerves per frame using semi-automated software ImageJ with NeuronJ (National Institutes of Health, Bethesda, MD). This parameter is arguably the most repeatable and provides maximum information about corneal nerve microstructure.[11] To note, the terms sub-basal nerve density and sub-basal nerve length measured as mm/mm2 are used interchangeably in the literature due to a lack of unanimous consensus in terminology.[4-8] ‘

Reviewer 2 Report

1. Authors wanted to demonstrate that the changes of corneal nerves could predict the development of oxaliplatin peripheral neuropathy. However, the study only shows that corneal nerve density modestly correlated with clinical peripheral neuropathy after 20 weeks of chemotherapy. The cross-sectional results cannot be inferred as predictive.

2. It is difficult to understand why the corneal nerve is correlated with only one time point.

3. Calculation of number of patients to include is mandatory.

4. Dry eye, a common side effect of Oxaliplatin, can also affect corneal nerve density. It should also be considered in the study.

5. The “Conclusions” section does not require references, just a concise and clear conclusion.

6. It is recommended to add the complete P-value to tables.

Author Response

  1. Authors wanted to demonstrate that the changes of corneal nerves could predict the development of oxaliplatin peripheral neuropathy. However, the study only shows that corneal nerve density modestly correlated with clinical peripheral neuropathy after 20 weeks of chemotherapy. The cross-sectional results cannot be inferred as predictive?

Reply: The results indeed present a modest association between corneal nerve change with peripheral neuropathy after 20 weeks of chemotherapy treatment. We agree that the cross-sectional results cannot be predictive, therefore, further studies are recommended in the manuscript to confirm these associations.

  1. It is difficult to understand why the corneal nerve is correlated with only a one-time point

Reply: We agree with the reviewer that it is important to correlate corneal nerve changes with other parameters as a function of time. The associations between corneal nerve density, corneal sensitivity, and total neuropathy score over the five time-points are discussed in the results section and table 3.

  1. Calculation of number of patients to include is mandatory?

Action: The following sentence is added in the methods section.

‘Annually approximately 200 patients are treated using oxaliplatin in Auckland Hospital, and the drug has been reported to cause sensory neuropathy in up to 80% of these patients.[12] Attempting to calculate a sample size based on repeated measures analysis was difficult due to the novel nature of the study. At present there is no estimate for the value of the mean corneal nerve density (the primary outcome measure) at each time point, nor is there a value for the correlation between each time point and these would be needed to calculate sample size. Hence, this research was defined as a longitudinal observational study without any need of sample size calculation. All the patients undergoing oxaliplatin treatment for gastrointestinal cancer who meet the criteria will be requested to participate. This observational study will gather the data and values needed to form the foundation for a detailed sample size calculation for future randomized trials.’

  1. Dry eye, a common side effect of Oxaliplatin, can also affect corneal nerve density. It should also be consideredin the study?

Action: The following sentence is added in the discussion section

‘Painful dry eye disease is associated with corneal nerve microstructural changes.[13]  Further research is needed to investigate the role of dry eye associated corneal nerve in patients undergoing chemotherapy.

  1. The “Conclusions” section does not require references, just a concise and clear conclusion.

Action: The following sentences are moved to the discussion section from the conclusion.

‘Oxaliplatin-induced peripheral neuropathy is related to the cumulative dose of oxaliplatin; the higher the cumulative dose, the greater the risk of developing neuropathy.[14] Unfortunately, the only effective method of preventing oxaliplatin-induced peripheral neuropathy is to reduce the dose of oxaliplatin delivered or terminate treatment.’

  1. In Table 2, what are the criteria for determining the increase and decrease of nerve density?  Do comparisons need to be made at the same follow-up time point?

Reply: The nerve density for all the patients were tracked at the same time points to ensure overall consistency. The rationale of Table 2 was to demonstrate the varied changes in corneal nerve density from baseline to their final appointment.

  1. It is recommended to add the complete P-value to tables.

Reply: We agree with the reviewer that p values are required for clearer understanding. Therefore, the p values for correlation analysis are stated in table 3. Table 1 details the presence of dendritic cells, neuromas, and neuropathy, alongside total neuropathy score and nerve conduction study (NCS) scores along with the number of patients with Relative Afferent Pupillary Defect (RAPD) and retinal abnormalities. The means of corneal nerve density and corneal sensitivity threshold at each visit with standard deviations are also noted. Stating the p-value in the table without an imputation analysis is statistically inaccurate with missing values at the final two time-points. The specific repeated measures analysis after employing the imputation model is added in the manuscript text.  Table 2 presents the tracking of individual corneal subbasal nerve density through the various appointments. 

Reviewer 3 Report

Dear authors,

Thank you far asking me to review this article. I found it well written. I only have a few comments.  

In discussion paragraph I would suggest to make some speculations regarding the reason for the not demonstrated association between oxaliplatin treatment and changes in corneal nerve density (since previous studies, as you have reported, did show this kind of association). 

Also,  I would mention the potential usefulness of assessing corneal sensitivity in patients undergoing chemotherapy. 

Author Response

  1. Thank you for asking me to review this article. I found it well written. I only have a few comments In discussion paragraph I would suggest to make some speculations regarding the reason for the not demonstrated association between oxaliplatin treatment and changes in corneal nerve density (since previous studies, as you have reported, did show this kind of association).

Reply: We thank the reviewer for the positive comments. As we have stated in the discussion section, previous studies either investigated cross-sectional changes in corneal nerves after chemotherapy in comparison with control participants (Chiang et al, 2021) or before and after three cycles of chemotherapy (Campagnolo et al 2013).  The cancer patients may have compromised corneal nerve microstructure before any treatment, therefore, unlike the previous studies, it is important to assess patients before, during, and after cessation of treatment to draw firm conclusions, as we did in our study. The discrepancies between the studies could potentially be due to the discontinuation of follow-up following chemotherapy cessation.

Discussion already states

‘Interestingly, corneal nerve fibre length and density were reduced in patients with upper gastrointestinal cancer without chemotherapy in comparison to age-matched controls, attributed to systemic peripheral neuropathy due to cancer.[15]

Action: The following sentence is added in the discussion section.

‘The inconsistencies between the studies could potentially be due to discontinuation of follow up following chemotherapy cessation.’

  1. Also,  I would mention the potential usefulness of assessing corneal sensitivity in patients undergoing chemotherapy

Action: The following sentence is added in the discussion section

‘There is possible merit in assessing corneal sensitivity threshold in patients undergoing chemotherapy to monitor peripheral neuropathy at various stages of their treatment.’

Round 2

Reviewer 2 Report

The cross-sectional study cannot be inferred as predictive, so the predictive description in the full manuscript should be modified. For example, "predicting the development of oxaliplatin peripheral neuropathy"(P40-P41).

Author Response

We are thankful for the reviewers’ constructive comment. We have made the requested changes as shown below in italics.

'In conclusion, corneal changes detected on confocal microscopy show moderate association with  peripheral neuropathy, indicating their potential to identify the development of oxaliplatin-induced peripheral neuropathy. However, further studies are required to confirm these findings.'

This manuscript is a resubmission of an earlier submission. The following is a list of the peer review reports and author responses from that submission.

Round 1

Reviewer 1 Report

This is an extremely interesting study on the use of in vivo confocal microscopy in patients receiving oxaliplatin for colorectal cancer. Very nice approach. Nevertheless, the conclusion is very harshly worded: “ … corneal changes detected on confocal microscopy appear unhelpful in predicting the development of oxaliplatin peripheral neuropathy”. The reviewer casts doubt on this because statistically nothing is said in the manuscript about the statistical stability/repeatability (1) in collecting and (2) analyzing images. To say this and at the same time to introduce and use a new method for analyzing image data (Oakley 2020 and McCarron 2021) is scientifically unacceptable.

Reviewer 2 Report

  1. In the introduction section, what are the effects of oxaliplatin chemotherapy on corneal nerves in previous literature?
  2. How to determine the sample size? Are the data sufficient for statistical analysis?
  3. What are the inclusion and exclusion criteria for patients?  Do you exclude people with existing eye diseases?
  4. In terms of the methods, are the patients treated at the same dose? How to judge the effect of dose on experimental results?
  5. How many confocal images were collected per patient?  How to access the representativeness of an image?
  6. Due to the high level of loss to follow-up, the average level at each visit point in Table 1 is not a good representative of the overall level.
  7. In Table 2, what are the criteria for determining the increase and decrease of nerve density?  Do comparisons need to be made at the same follow-up time point?
  8. Is it necessary to show OCT images?
  9. Due to loss of follow-up and small sample size, this study has a noteworthy error bias.